# Gut–Brain Axis: Insights from Hippocampal Neurogenesis and Brain Tumor Development in a Mouse Model of Experimental Colitis Induced by Dextran Sodium Sulfate

**DOI:** 10.3390/ijms231911495

**Published:** 2022-09-29

**Authors:** Roberta Vitali, Clara Prioreschi, Laura Lorenzo Rebenaque, Eleonora Colantoni, Daniela Giovannini, Sarah Frusciante, Gianfranco Diretto, Francisco Marco-Jiménez, Mariateresa Mancuso, Arianna Casciati, Simonetta Pazzaglia

**Affiliations:** 1Biomedical Technologies Laboratory, Agenzia Nazionale per le Nuove Tecnologie, l’Energia e lo Sviluppo Economico Sostenibile (ENEA), 00123 Rome, Italy; 2Departamento Producción y Sanidad Animal, Salud Pública Veterinaria y Ciencia y Tecnología de los Alimentos, Universidad CEU-Cardenal Herrera, 46115 Valencia, Spain; 3Biotechnology Laboratory, l’Energia e lo Sviluppo Economico Sostenibile (ENEA), 00123 Rome, Italy; 4Laboratory of Biotechnology of Reproduction, Institute for Animal Science and Technology (ICTA), Universitat Politècnica de València, 46022 Valencia, Spain

**Keywords:** DSS-induced colitis, brain, neuroinflammation, adult hippocampal neurogenesis, metabolomics, medulloblastoma

## Abstract

Chronic inflammatory bowel disorders (IBD) are idiopathic diseases associated with altered intestinal permeability, which in turn causes an exaggerated immune response to enteric antigens in a genetically susceptible host. A rise in psych cognitive disorders, such as anxiety and depression, has been observed in IBD patients. We here report investigations on a model of chemically induced experimental colitis by oral administration of sodium dextran sulfate (DSS) in C57BL/6 mice. We investigate, in vivo, the crosstalk between the intestine and the brain, evaluating the consequences of intestinal inflammation on neuroinflammation and hippocampal adult neurogenesis. By using different DSS administration strategies, we are able to induce acute or chronic colitis, simulating clinical characteristics observed in IBD patients. Body weight loss, colon shortening, alterations of the intestinal mucosa and fecal metabolic changes in amino acids-, lipid- and thiamine-related pathways are observed in colitis. The activation of inflammatory processes in the colon is confirmed by macrophage infiltration and increased expression of the proinflammatory cytokine and oxidative stress marker (*Il*-*6* and *iNOS*). Interestingly, in the hippocampus of acutely DSS-treated mice, we report the upregulation of inflammatory-related genes (*Il*-*6*, *Il*-*1β*, *S*-*100*, *Tgf*-*β* and *Smad*-*3*), together with microgliosis. Chronic DSS treatment also resulted in neuroinflammation in the hippocampus, indicated by astrocyte activation. Evaluation of stage-specific neurogenesis markers reveals deficits in the dentate gyrus after acute and chronic DSS treatments, indicative of defective adult hippocampal neurogenesis. Finally, based on a possible causal relationship between gut-related inflammation and brain cancer, we investigate the impact of DSS-induced colitis on oncogenesis, using the *Ptch1^+/−^*/C57BL/6 mice, a well-established medulloblastoma (MB) mouse model, finding no differences in MB development between untreated and DSS-treated mice. In conclusion, in our experimental model, the intestinal inflammation associated with acute and chronic colitis markedly influences brain homeostasis, impairing hippocampal neurogenesis but not MB oncogenesis.

## 1. Introduction

The gut–brain axis, formed by blood and lymphatic vessels, paves the way for microbiota to impact the brain. Inflammatory bowel diseases (IBDs) are chronic intestinal disorders frequently associated with comorbid psychological and cognitive deficits [1]. For example, it has recently been reported that IBD patients exhibit double the rate of depression compared to healthy controls, with this prevalence being heightened during active IBD [2]. Nevertheless, the mechanism(s) through which these extraintestinal deficits develop remain largely unknown. IBD also dramatically increases the risk of developing intestinal cancer. In fact, malignancies are the second most common cause of death after cardiovascular diseases in both sexes of IBD patients [3]. Patients with IBD also have an increased risk of developing a variety of extraintestinal malignancies. Longstanding inflammation and the degree of immunosuppression as a result of IBD treatment appear to be the main driving factors for IBD-related carcinogenesis [3].

Relatively few IBD mouse models have been developed, with each able to replicate many, but not all, aspects of human disease. The dextran sodium sulfate (DSS) model of chemically induced colitis in mice is a robust model of either acute or chronic colonic inflammation depending on administration strategies, with similar disease features seen in ulcerative colitis patients, including diarrhea, body weight loss, mucosal ulcers, shortened colon and dysbiosis, as well as behavioral alteration [4,5]. Recently, the influence of experimental colitis induced by DSS on adult neurogenesis was investigated [6,7,8,9], and either acute or chronic colitis was shown to impair neurogenesis in the dentate gyrus (DG).

Gut dysbiosis is implied in the dysfunctional gut–brain axis, with specific classes of metabolites also triggering IBD [10]. Furthermore, alterations in the microbiome metabolism have been associated with the development of different human neurodegenerative and neuropsychiatric disorders. Metabolomics approaches could thus help to elucidate the pathogenesis of this complex disease, also potentially improving its understanding and management or identifying specific biochemical markers [11,12]. To explore the metabolite role in IBD pathogenesis, we here compared the metabolite changes in a fecal sample from acute and chronic DSS-treated mice vs. untreated mice.

The present in vivo study aims at exploring the crosstalk between the intestine and the brain in a DSS-induced colitis mouse model, evaluating the consequences of cancer and non-cancer brain pathologies. Specifically, we have investigated the effects of intestinal inflammatory processes (i) on neuroinflammation and hippocampal neurogenesis in C57BL/6 mice and (ii) on tumorigenesis in the *Patched1* heterozygous mice (*Ptch1^+/−^*/C57BL/6), predisposed to the development of medulloblastoma (MB), a cerebellar tumor, representing the most common malignant pediatric brain cancer [13,14]. MB likely arises from the external granular layer (EGL) of the cerebellum. Interestingly, the EGL of the cerebellum and the DG of the hippocampus are both mainly constituted by granule neurons. These brain structures share long-lasting developmental features, as they are still developing in the last trimester of human gestation and a similar development is taking place during the mouse perinatal period, extending their vulnerability to toxic insults affecting neurogenesis and synaptogenesis.

To simulate the clinical characteristics featuring the acute and chronic phases of colitis in IBD patients, intermittent DSS administration strategies, consisting of the addition of DSS to the drinking water of different durations, were adopted. A significant percentage of weight loss and colon shortening were observed for both acute and chronic colitis, together with the presence of intestinal mucosa alterations. In fecal samples of DSS-treated mice, we also detected marked metabolic changes—converging in pathways regulating (i) amino acids (alanine, aspartate, glutamate and phenylalanine), (ii) different classes of lipids and (iii) the thiamine metabolism—and representing disease signatures for bowel syndrome and Crohn’s disease. Macrophage infiltration and increased expression of proinflammatory cytokines *Il*-*6* and *iNOS* in the colon were indicative of the activation of an inflammatory process. Notably, significantly increased expression of *Il*-*6*, *Il*-*1β*, *S*-*100*, *Tgf*-*β* and *Smad*-*3* was also detected in the hippocampus of acutely DSS-treated mice, together with microglia activation, indicative of neuroinflammation. Chronic DSS treatment also resulted in increased expression of markers of astrocyte activation (*Smad*-*3* and *S*-*100*), supportive of neuroinflammation in the hippocampus. Evaluation of stage-specific neurogenesis markers showed alteration in different cell populations in the DG after acute and chronic DSS treatments, revealing defective adult hippocampal neurogenesis. Conversely, oncogenesis in *Ptch1^+/−^*/C57BL/6 mice indicated no differences in MB tumor development between untreated and DSS-treated mice. In conclusion, our study demonstrates that inflammation associated with acute and chronic colitis results in altered neural function, impairing hippocampal neurogenesis but did not affect MB oncogenesis in our experimental settings. These results may have important implications in the evaluation of the risk of IBD-related extra-intestinal cancer and non-cancer pathologies.

## 2. Results

### 2.1. Schematic Experimental Design

The overall experimental scheme for the induction of acute and chronic colitis by DSS is illustrated in Figure 1. A pilot study has been carried out to identify the appropriate concentration of DSS (Appendix A). For the induction of acute DSS-induced colitis, C57BL/6 male mice of 8 weeks of age were treated for 7 days with 0.7% of DSS and sacrificed after 3 or 7 days for molecular and histological analyses of the colon and hippocampus (Figure 1A). For the induction of chronic DSS colitis, C57BL/6 male mice at 8 weeks of age were exposed to three repeated DSS cycles (7 days each followed by a recovery period of 14 days with tap water); 1 week after the end of the third DSS cycle, the mice were sacrificed and colon and brain were collected for molecular and histological analyses (Figure 1B). Fecal samples were collected and snap frozen at the times indicated in Figure 1A and 1B. For the tumorigenesis experiment, *Ptch1^+/−^*/C57BL/6 mice of mixed gender were treated with an extended DSS protocol consisting of 5 DSS cycles followed by 14 days of recovery with tap water, to cover the majority of the tumor developmental window (Figure 1C). Mice were then monitored for MB development for the whole lifespan.

### 2.2. Characterization of Acute DSS-Induced Colitis

The severity of colitis was assessed by evaluating (i) body weight, (ii) colon length, (iii) histological score and (iv) expression of inflammatory-related markers. While the untreated mice gained weight during the treatment, the DSS group progressively lost weight, with the minimum peaking at 9 days from the start of the treatment. In particular, on day 9, DSS-treated mice showed a loss in body weight of −7.75% ± 2.76 with respect to their initial weight, while untreated mice showed a gain of +6.24% ± 1.46 (*p* = 0.0006, Figure 2A). We also detected a significant reduction in the colon length of 7% in the DSS-treated compared to untreated mice (DSS: 8.12 cm ± 0.16 vs. UN: 8.73 cm ± 0.23, *p* = 0.0368; Figure 2B). We next evaluated the histopathological score, taking into consideration the increase in cellularity (lymphocytes and neutrophils in the lamina propria), edema, erosions or ulcerations of mucosa and loss of mucosal architecture [15], finding a significant increase in the histological score in DSS-treated mice (UN: 0.21 ± 0.10 vs. DSS: 2.60 ± 0.40, *p* < 0.0001; Figure 2C,D). In strong agreement with the above findings, the mRNA expression analysis of *Il*-*6* and *iNOS* in the intestinal mucosa was significantly increased (*p* = 0.0159 and *p* = 0.0381, Figure 2E,G), but not for *Il*-*1β* (Figure 2F) in DSS-treated mice. Furthermore, immunohistochemistry analysis for the microglial marker ionized calcium-binding adapter molecule 1 (Iba-1) revealed its activation (*p* = 0.0080) in DSS-treated mice (Figure 2H,I). Altogether, body weight loss, colon shortening and histopathological and molecular signs of crypt damage and intestinal inflammation demonstrated the presence of acute colitis in experimentally DSS-treated mice.

### 2.3. Characterization of Chronic DSS-Induced Colitis

Similarly to the acute DSS treatment, the maximum weight loss percentage was observed on day 9 of the first DSS cycle, when DSS-treated mice showed a significant reduction of body weight of −12.4% with respect to their initial weight, while the untreated mice showed a body weight increase of +7.18% (DSS: −12.37 ± 2.97 g vs. UN: ±7.18 ± 1.15 g, *p* < 0.0001, Figure 3A). A statistically significant shortening of the colon length of 16.4% in DSS compared to the untreated group (DSS: 7.75 ± 0.25 cm vs. UN: 9.27 ± 0.11 cm, *p* < 0.0001, Figure 3B) was also observed. Compared to untreated mice, chronic DSS treatment induced a highly significant increase in the fibrotic score, assessed in Masson’s trichrome stained tissue sections through evaluation of the amount of collagen in the intestinal mucosa and percentage of tissue involved (DSS: 7.61 ± 0.33 vs. UN: 0.75 ± 0.30, *p* < 0.0001; Figure 3C,D). Finally, analysis of gene expression in the intestinal mucosa by PCR showed a statistically significant increase in the expression of *Il*-*6* (*p* = 0.0003) and *iNOS* (*p* = 0.049) (Figure 3E,G), but not of *Il*-*1β* (Figure 3F) in DSS-treated mice. In the complex, all macroscopic (body weight and colon length), histopathological (crypt damage and presence of fibrosis) and expression patterns of inflammatory-related genes indicated the presence of chronic colitis in these mice.

### 2.4. Comparative Study of the Metabolomes in Fecal Samples from DSS-Treated and Untreated Mice

The fecal metabolomic profiles of acute colitis DSS-treated mice (on days 7 and 14 from the start of the treatment) and chronic colitis DSS-treated mice were compared with untreated mice in order to unravel the metabolite role in IBD pathogenesis. We carried out an untargeted metabolomics analysis to gain a general overview of the metabolic changes occurring. Principal component analysis (PCA) biplots demonstrated clustering among the DSS-treated groups, and distant clusters were observed for untreated mice (Figure 4A). This was correlated with hierarchical clustering of the different samples, demonstrating that deviations induced by colitis were higher than those related to mouse aging (9 weeks vs. 16 weeks old animals). The metabolic profiling yielded a total of 839 metabolites. For acute colitis, the *t*-test analysis detected 156 and 136 differentially accumulated metabolites (DAMs) for 7 and 14 DSS-treated compared with untreated mice, respectively (*p* < 0.05), indicating a trend of recovery at the end of the DSS treatment (Figure 4B,C). For chronic colitis, the number of DAMs was 481, compared with untreated mice (*p* < 0.05), indicating increasing metabolic alteration (Figure 4B,C). Only 48 of the DAMs for acute and chronic colitis have been identified in the Metabolite database and are reported in Table 1. Moreover, a general down-accumulation of the significant metabolites was recorded in all the DSS-treated groups (Figure 4C). More specifically, Venn diagrams of the DAMs showed that 27 of the DAMs were common between chronic and acute conditions (Figure 4B), and all were down-accumulated metabolites. The chemical natures of the identified DAMs in chronic colitis were the following: organic acids, nucleic acids, organoheterocyclic compounds, polyketides, benzenoids and different classes of lipids (prenol lipids, fatty acils, sterol lipids and glycerophospholipids) (Table 1). In addition, pathway analysis of the DAMs in chronic colitis, performed by using the Metaboanalyst tool coupled to a manual curation step, allowed for identifying the following four dysregulated pathways: (i) alanine, aspartate and glutamate metabolism, (ii) thiamine metabolism, (iii) lipid and (iv) phenylalanine metabolism. In fact, the metabolites related to these pathways, D-aspartic acid, thiamine, lipids belonging to different sub-classes and phenylethylamine, respectively, were down-regulated in chronic DSS (*p* < 0.05). Instead, in acute colitis, no pathway was enriched.

To evaluate metabolic changes involved in colitis chronicization, we also compared the metabolic profiles of acutely (at 14 days) and chronically DSS-treated mice, in which fecal samples were collected 1 week after the stop of DSS treatment. In DSS-chronically treated mice, we detected 101 DAMs, 84 of which were down-regulated and 17 upregulated (*p* < 0.05). Among the identified structures of the DAMs were polyketides (Afrormosin-like and Bergaptol-like), sterol lipids (bile acids as Lithocholic acid glycine conjugate; vitamin D-like 1α.25-dihydroxy-26.27-dimethyl-20.21-methano-23-oxavitamin D3/1α.25-dihydroxy-26.27-dimethyl-20.21-methano-23-oxacholecalciferol; steroids as 16alpha.17-Isopropylidenedioxy-6alpha-methylprogesterone and 11b-hydroxyproge-sterone), and glycerophospholipids (LysoPC(18:1(9Z)). In addition, a series of lipids and lipid-like molecules were also strongly down-accumulated, including vitamin A (3,4-didehydro-retinoate and 9-cis-Retinoic acid) and vitamin E compounds (alpha-Tocotrienol), fatty acils and their derivatives (pimelic, suberic and azelaic acids, and 15-Deoxy-Δ12.14-prostaglandin J2-2-glycerol ester, oleamide and octadecanamide). Nevertheless, pathway analysis of the identified DAMs did not identify significantly enriched pathways.

Overall, fecal metabolic profiles indicate increasing metabolic alterations with a reduced abundance of metabolites in chronic vs. acute colitis.

### 2.5. Acute and Chronic Colitis Enhanced Inflammatory Response in the Hippocampus with a Distinct Cellular Pattern

To evaluate the influence of gut colitis on neuroinflammation, we investigated the microglia and astrocyte cell populations in the hippocampus after the induction of acute and chronic gut colitis by DSS treatment.

Brain sections of mice acutely or chronically treated with DSS were immune-stained for Iba-1, a microglial marker (Figure 5A). The evaluation of the number of cells labeled by Iba-1 in the whole hippocampus showed a significant increase (+29.2%) in DSS-acutely treated mice (*p* = 0.042, Figure 5B) but not in DSS-chronically treated mice (Figure 5C). Instead, compared to control mice (UN), the hippocampus of DSS-chronically treated mice exhibited a significant increase in the number of astrocytes labeled by Gfap (+34.3%) (*p* = 0.050, Figure 5D,F) that was not observed in DSS acutely treated mice (Figure 5E). On the whole, these data indicated that microglia activation in the hippocampus was an early response to DSS acute treatment, replaced later by the induction of astrogliosis in DSS chronically treated mice.

### 2.6. Acute and Chronic Colitis Enhanced Inflammatory Response in the Hippocampus with a Distinct Cytokine Profile

Cytokines are principal mediators of the innate and adaptive immune responses in mucosal inflammation [9]. To evaluate the influence of gut inflammation on the brain, we assessed the inflammatory status in the manually dissected hippocampus of acutely and chronically DSS-treated mice (Figure 6). Compared to the control group (UN), the analysis showed a significant increase of *S*-*100* (over 100-fold, *p* = 0.023, Figure 6A), *Il*-*6* (2-fold, *p* = 0.015, Figure 6B), *Il*-*1β* (9-fold, *p* = 0.009, Figure 6C), *Tgf*-*β* (9-fold, *p* = 0.024, Figure 6D) and its downstream effector *Smad*-*3* (5-fold, *p* = 0.02, Figure 6E), suggesting that gut inflammation propagated to the brain and that gut and hippocampus shared a very similar inflammatory pattern at early time after DSS treatments.

Notably, in accordance with the increase in astrocytes in chronically treated mice, we found a significantly increased expression of *S*-*100*, a specific marker for astrocytes, (4.5-fold, *p* = 0.0043, Figure 6F) compared to untreated mice. Although the expression of *Il*-*6*, *Il*-*1β* and *Tgf*-*β* mRNA was not different between DSS-treated and untreated mice (Figure 6 G-I), the expression of the downstream *Tgf*-*β* pathway factor *Smad*-*3*, was significantly upregulated (2-fold, *p* = 0.03, Figure 6J), suggesting a chronic activation of the *Tgf*-*β* signaling pathway in the hippocampal region of DSS chronically treated mice. Altogether, the pattern of cytokines and other inflammatory-related genes showed stronger alterations in the hippocampus during the acute phase of colitis.

### 2.7. Acute and Chronic Colitis Induced Different Stage-Specific Changes in Hippocampal Neurogenesis

To examine whether the presence of neuroinflammation and astrogliosis in the hippocampus of mice with DSS-induced colitis also affects neurogenesis, we evaluated the cellular composition of the subgranular (SGZ) and granular zones of DG. The different cell populations in the SGZ can be distinguished based on a combination of cellular morphological and immunohistochemical characteristics, using the stage-specific cellular markers of neurogenesis (Figure 7A).

Adult hippocampal neurogenesis is a multi-stage process originating from slowly dividing/quiescent radial glia-like stem cells (RGL), with astrocytic properties (Gfap^+^), characterized by a single radial process that extends through the granular cell layer of the DG. They give rise to small round transit-amplifying intermediate progenitor cells (Sox2^+^ and Ki67^+^) that then acquire a neuronal phenotype (Dcx^+^) and finally integrate into the existing neuronal networks (summarized in Figure 7A). Compared to untreated samples, acute-DSS treatment induced a significant depletion of RGL cells (−34.7%, *p* = 0.042, Figure 7B) and Sox2-labeled transit amplifying cells (−22.08%, *p* = 0.040, Figure 7D). However, no change was observed in the number of Ki67-proliferating or Dcx-labeled immature neurons (Figure 7C,E). On the contrary, DSS-chronic treatment only induced a significant decrease in the number of Ki67+ proliferating cells (−49.0%, *p* = 0.0032) (Figure 7G), while no alterations in the number of RGL stem cells (Figure 7F), Sox2-transit amplifying cells and Dcx-immature neurons were observed (Figure 7H,I). Overall, our data showed that both acute and chronic DSS-treatment produced alterations in cell populations of adult DG with greater staminal/progenitor characteristics such as Gfap, Sox2 and Ki67 that were in fact more responsive to intestinal inflammation than Dcx neuronal populations.

### 2.8. Relationship between Gut Inflammation and Brain Tumorigenesis in Ptch1^+/−^ Mice

Patients with IBD have been shown to be at increased risk of developing extra-intestinal malignancies, especially those who have long-lasting immunosuppressive therapies [3]. To help establish the role of the IBD in the development of extra intestinal malignancies—eliminating the impact of immunosuppressive drugs that may represent a confounding risk factor in IBD patients—we took advantage of the *Ptch1^+/−^* heterozygous knockout mice, a well-characterized MB model recapitulating many of the histological and molecular features of the human tumor counterpart [16]. MB, the most common malignant nervous system tumor in childhood, is thought to arise from disruptions in cerebellar development [17]. Given that the majority of MB development in *Ptch1^+/−^*/C57BL/6 mice occurs between 10 and 22 weeks of mouse age [18], to cover with DSS treatment most of the window of tumor development, we adopted an extended DSS protocol consisting of 5 repeated cycles, each alternated with 2 recovery weeks with tap water (Figure 1C), with the DSS treatment spanning from 8 to 21 weeks of mouse age. As a first step, we confirmed the induction of chronic colitis by DSS-treatment in the colon of *Ptch1^+/−^* mice (Figure 8A) and, similarly to DSS-treated C57BL/6 WT mice, we found a significant increase of Gfap (+102.69%, *p* = 0.0052, Figure 8B,C), indicative of astrogliosis in the hippocampus of the *Ptch1^+/−^* mice but not in the cerebellum (Figure 8D,E).

Next, we compared MB tumorigenesis in DSS-treated and untreated *Ptch1^+/−^* mice, finding no differences in the rate of tumor development between the two groups of mice (Figure 8F,G). Concordantly, tissue sections of MBs developed in DSS-treated *Ptch1^+/−^* mice during the treatment interval or in age-matched untreated *Ptch1^+/−^* mice, showed no differences in expression levels of Gfap and Ki67 after immunostaining (Figure 8H,J). Overall, these data demonstrated the absence of neuroinflammation in the cerebellum of mice with DSS-induced chronic colitis and the consequent lack of effect of DSS treatment on MB oncogenesis.

## 3. Discussion

A recent study found that, compared to healthy individuals, IBD patients were twice as likely to suffer from psychiatric disorders such as depression and anxiety, although the mechanisms linking intestinal inflammation and depression-like behavior are still largely unknown [2]. They often develop obsessive-compulsive disorders, panic disorders, depression and anxiety [19,20,21,22]. Furthermore, IBD is associated with an overall increased risk of developing extra-intestinal cancer [3]. In this study, we wanted to investigate the potential effects of colitis-drive inflammation on the brain, examining the influence on neuroinflammation, hippocampal neurogenesis and brain tumor development. By adopting different DSS administration strategies, we have experimentally reproduced acute and chronic phases of DSS-induced intestinal inflammation that closely feature the course of IBD in humans, characterized by acute disease followed by remission.

We here demonstrate that acute colitis causes loss of body weight, colon shortening, intestinal damage and high levels of inflammation with elevated local levels of *Il*-*6* and *iNOS*, as well as microglia activation in the colon. Interestingly, gut inflammation has been implicated in endothelial cell dysfunction, facilitating the translation of proinflammatory mediators from the gut to the peripheral circulation [23,24] and, in fact, increased expression of many immune factors has been reported in the serum of IBD patients [25,26]. Moreover, we identified marked metabolomic alterations in both acute and chronic induced colitis mouse models, with nearly the totality of down DAMs. Furthermore, more pronounced metabolic changes were detected in chronic than acute colitis animals. This is in agreement with previous findings showing the occurrence of a high extent of metabolic changes along with the progression of acute to chronic colitis in DSS-treated mice [27], thus suggesting that disease development has profound implications for the cell metabolome. Some alterations were expected and could be considered as IBD biochemical markers and positive controls in our study as follows: this is the case of molecules involved in vitamin A, D and E metabolism, which are known to undergo serious deficiency status conditions [28,29,30], as well as polyunsaturated fatty acids (PUFAs)/eicosanoids, which have been reported to exert a strong anti-inflammatory effect [31,32].

The identified DAMs and corresponding pathways analysis in chronic colitis mice highlighted a set of common alterations in (i) alanine, aspartate and glutamate metabolism, (ii) thiamine metabolism, (iii) lipid and (iv) phenylalanine metabolism. Most of these compounds represent non-dispensable primary biosynthetic routes for cell metabolism, and their strong alteration provides a deeper understanding of metabolic changes in IBD. More in general, nutritional deficiencies in IBD result from reduced intake, malabsorption and systemic inflammation due to active diseases [33,34]. In this context, nutritional therapy, including amino acids and thiamine (vitamin B1), is frequently used to overcome nutrient deficiencies. In particular, thiamine deficiency is relatively common in IBD, leading to fatigue, irritability, poor memory, sleep disturbances, nausea and vomiting, and abdominal pain, as well as severe neurologic deficits such as Wernicke’s encephalopathy [35,36]. Noteworthy that DAMs induced in our experimental model of colitis were related to four disease signatures, including irritable bowel syndrome, Autism, Crohn’s disease, ulcerative colitis and colorectal cancer. The metabolites related with these diseases were D-Aspartic acid for irritable bowel syndrome, Autism and Crohn’s disease, phenylethylamine for Crohn’s disease and ulcerative colitis, and D-aspartic acid, azelaic acid, suberic acid, Glycyl-Phenylalanine, 5-(2-Hydroxyethyl)-4-methylthiazole, 9Z-Eicosenoic acid, hexadecanedioic acid mono-L-carnitine ester, octadecanamide, thiamine and pimelic acid for Colorectal cancer, all of them down-regulated in DSS groups.

Interestingly, a large number of pathophysiological evidences described hereafter, might link colitis-related thiamin deficiency to IBD brain manifestations. Thiamin is involved in an organism’s energy metabolism, processing carbohydrates into energy. In the human body, thiamine-rich tissues include the brain, among others [37]. Thiamine deficiency contributes to a number of conditions spanning from mild neurological and psychiatric symptoms (confusion, reduced memory and sleep disturbances) to severe encephalopathy [38]. Noteworthy, thiamine deficiency has been reported to generate neurological manifestations [39], involving neurodegeneration [40]. Several studies have also shown an inverse association between thiamine levels and depression [41,42], which is reversed by administering thiamine [43]. Finally, impairment of hippocampus-dependent memory formation and loss of hippocampal neurons and dendritic spines were reported in thiamin-deficient mice [44], strengthening the link between altered thiamin metabolism identified by our pathway analysis and hippocampal alteration in mice with colitis.

Our analysis revealed notable changes in amino acid metabolism. Together with aspartate, glutamate is the major excitatory neurotransmitter in the brain. Dysfunctional brain energy metabolism is implicated in several neurodegenerative diseases [45,46,47,48], and cellular glutamate metabolism is a growing area of interest in neurodegenerative disease research [49,50]. In strong analogy with our results, a comparative analysis of fecal metabolome features among multiple depressive animal models identified significantly altered pathways belonging to lipids and amino acid metabolism, suggesting that these metabolic alterations may be linked to IBD-related depression [51].

Cytokines are key mediators of cellular interactions in the intestine in both physiology and pathophysiology. Cytokine functions are context-dependent and can exert opposing effects depending on the stage of inflammation. The organized cellular network maintains intestinal homeostasis by physically excluding commensal microbes from penetrating host tissue and actively promoting host defense and immune regulation [52]. The single layer of intestinal epithelial cells in the gut not only represents the physical barrier separating the microbiota from the mucosa but also integrates incoming signals from commensals, pathogens and dietary components [53].

Mechanistically, IBD-related proinflammatory cytokines have been reported to diminish blood–brain barrier (BBB) integrity and enhance its permeability. For instance, a reduction in the tight junction proteins occludin and claudin-5 has been reported in the hippocampus of DSS-treated mice due to elevated *Il*-*6* levels [54]. Cytokines and proinflammatory genes may, thus, enter into the brain, locally modulating the microglia [6,54,55]. Activated microglia produces and secretes a plethora of neurotoxic proinflammatory mediators including cytokines (e.g., *Il*-*1β*, *Il6* and *Tgf*-*β*) negatively modulating hippocampal neurogenesis [8]. Noteworthy, in our experimental model, during acute colitis, we report in the hippocampus (i) neuroinflammation, detectable as increased microglia, (ii) upregulation of mRNA expression of proinflammatory cytokines including *Il*-*1β*, *Il*-*6* and *Tgf*-*β* and (iii) disturbance of neurogenesis with depletion of stem/progenitor cell compartments (Gfap and Sox2).

Adult hippocampal neurogenesis mainly occurs in the SGZ in the DG of the hippocampus, resulting in the formation of new granule cells from neural progenitor cells (NPCs). Albeit mainly quiescent, neurogenesis persists throughout adulthood in the mammalian brain. The hippocampus plays an important role in sustaining normal memory and cognition, and perturbations of neurogenesis are known to contribute to neurological impairment. The proliferation, maturation and survival of newborn neurons, as well as their incorporation into the hippocampal neuronal network, are determined by multiple factors. A great deal of work has been devoted to dissecting the role of cytokines in the regulation of neurogenesis.

Our data show a significant decrease in the stem/progenitor cell compartments of the DG (Gfap+ and Sox2+ cells) in mice with active colitis. This may be explained by the observation that NPCs constitutively express receptors for proinflammatory cytokines [56] and that the establishment of a proinflammatory environment in the neural stem cell (NSC) niche is detrimental towards neurogenic activities. *Il*-*1β* and *Il*-*6,* proinflammatory cytokines upregulated in the hippocampus of our mice with acute colitis, are potent suppressors of neurogenesis [57,58,59]. The release of inflammatory cytokines in the hippocampus in response to peripheral-derived factors may be triggered by microglia activation. Interestingly, our data support the hypothesis of local production of cytokines upon gut-derived signals, as besides increased microglia, in the hippocampus of acutely DSS-treated mice, we observed a significant *Il*-*1β* upregulation that was not observed in the colon. Noteworthy, inflammatory processes are involved in the pathophysiology of depression [60,61,62] and microglia dysfunction is considered a key event in depression, with patients showing increased blood concentration of proinflammatory cytokines such as IL-1β, IL-6 and TNF-α [63,64].

IBD is a chronic condition characterized by acute exacerbations followed by remissions. Under normal conditions, a proinflammatory initial phase elicited by infection or damage is followed by pathogen clearance and tissue repair, which ultimately restores homeostasis. Failure to resolve inflammation, however, results in chronic inflammation and uncontrolled tissue remodeling, such as fibrosis [65], that might occur independently of classic proinflammatory mediators. We here have compared the hippocampal effects of acute and chronic colitis, showing a more severe deregulation pattern of cytokines during the acute vs. chronic colitis phase, when the expression levels of *Il*-*6* and *Il*-*1β* did not differ anymore between chronically treated and untreated mice. In addition, during chronic colitis, hippocampal microgliosis, initially detected as increased Iba-1 staining in the acute phase, was replaced by astrogliosis and DSS chronically treated mice overexpressed *Gfap* and the astrocytic marker *S*-*100* in their hippocampus. It is likely that astrocytosis in the hippocampus might be dependent on increased microglia. Microglia are known to modulate proinflammatory and neurotoxic activities in astrocytes, [66] and microglia-derived Il-1β is a well-known inducer of astrocytosis [67]. Reactive astrocytosis is common after brain injury and disease; however, its functions have been debated, with previous studies showing they both hinder and support central nervous system (CNS) recovery [1,68,69]. Therefore, it has not been clear under what contexts they may be helpful or harmful. During chronic colitis, also the deregulation of *Tgf*-*β* signaling was partially restored compared to acutely DSS-treated mice as indicated by lack of *Tgf*-*β* upregulation and milder upregulation of *Smad*-*3* in chronically DSS-treated compared to untreated mice. Finally, in full accordance with the higher rate of depression-like symptoms during the acute IBD phase [20], the decline of neurogenesis was milder during chronic compared to acute colitis, when only the cell compartment labeled by Ki67 was significantly decreased while alterations of the stem/progenitor cell compartment were resolved.

The influence of experimental colitis induced by DSS on adult neurogenesis has been recently investigated in other studies [6,7,8,9]. DSS-treated mice also exhibited reduced adult hippocampal neurogenesis and increased activation of microglia and astrocytes [7]. More recently, acute colitis by DSS was reported to increase neurogenesis, though animals with chronic colitis had a normal level of neurogenesis, the newborn neurons showed deficits in their integration into the functional circuitry [9]. Finally, following induction of colitis at weaning (P21) via administration of low-dose DSS that resulted in dysbiosis and colonic inflammation, adult mice (P56) displayed brain/behavioral deficits [8].

The lack of available data on human adult neurogenesis in patients with chronic peripheral inflammatory disease such as IBD, due to the impossibility of carrying out labeling with neural stage-specific markers in humans, makes results of animal studies precious evidence of the link between neurogenesis disturbance and IBD. The association between reduced neurogenesis and depression in humans can only be inferred by the reduced hippocampal volume in depressed individuals [70]. In addition, proton magnetic resonance spectroscopy has shown changes in brain structure [71] as well as evidence for abnormal hippocampal glutamatergic neurotransmission in IBD patients [72]. Instead, experimental IBD models showing a clear connection between intestinal inflammation and changes in brain function, represent a valuable tool to investigate the mechanisms of extra-intestinal IBD manifestations, especially in the brain, as well as therapeutic strategies for prevention. However, relatively few experimental studies thus far have employed experimental colitis to investigate the influence of the inflammatory status related to gut inflammation on hippocampal neurogenesis [6,7,8,9] and none, to our knowledge, on brain tumor development.

Notably, increased risk for extraintestinal malignancies such as non-Hodgkin lymphomas and skin cancers has been reported in IBD patients [3] and a possible impact of the gut–brain axis on the immune system state, inflammation and brain tumorigenesis has also been proposed [73]. In humans, the proinflammatory cytokines IL-1β, IL-6, IL-8, IL-12, GM-CSF and TNF-α have been involved in brain tumor initiation and progression [74]. In glioblastoma, for instance, IL-6 has been reported to induce NFκB, resulting in STAT3 activation and greater tumor aggressiveness [75], while inhibition of IL-6/STAT3 and NFκB decreases glioma growth [76], and these results support an impact of inflammation on brain tumor development.

In this study, we focused on MB, a heterogeneous cerebellar tumor characterized by molecular subgroups with distinct gene expression profiles. Activation of the SHH signaling pathway characterizes one of these molecular subgroups, comprising about 30% of total MBs in humans. A common driver for this subgroup includes mutations or deletions of the negative pathway regulator *PTCH1* [77]. We here employed a mouse model with heterozygous deletion of the *Ptch1* gene that recapitulates many characteristics of the human MB. Notably, our data show that induction of experimental colitis by DSS does not modify the rate of MB development in *Ptch1^+/−^* mice. Furthermore, while we noted induction of neuroinflammation in the hippocampus of *Ptch1^+/−^* mice following DSS treatment, no signs of neuroinflammation were detectable in the cerebellum, suggesting that local regional differences in brain susceptibility to the signals driven by DSS-induced gut inflammation might be responsible for the lack of effect on MB tumorigenesis. Changes in each region of the brain caused by DSS-induced colitis were not consistent and varied depending on location. Regional brain differences in the peripheral inflammatory response caused by DSS experimental colitis have been reported in another study [78], suggesting that region-specific studies are needed to understand the precise mechanisms by which chronic intestinal inflammation affects the brain.

An aspect worth considering is that Shh-dependent MB has been reported to arise from Sox2+ granule neuron precursors in the EGL cells of the cerebellum. The appearance of MB precursor lesions over the surface of the *Ptch1^+/−^* mouse cerebellum [79] is associated with a differentiation failure of transient Sox2+ EGL cells, presenting a clear mechanism for tumor initiation as the aberrant persistence of a progenitor cell state normally lost during the development of the cerebellum [80]. Normal granule neuron development in the cerebellum is rooted in a Sox2+ cell-driven hierarchy, with Sox2+ cells becoming the first newborn neurons Dcx+ and finally differentiating into NeuN+ cells. Precursor MB lesions, as well as full MB tumors, consist of a mixed population of Sox2+, Dcx+ and NeuN+ cells, suggesting the persistence of developmental processes as mechanisms for disease initiation and progression consisting of dysregulation of hierarchical growth that is characteristic of many cancer types [81]. Given that both the cerebellum and hippocampus are mostly constituted by granule neurons, it is not surprising that the Sox2+ cell-driven hierarchy in cerebellar granule neuron precursors closely resembles the steps of maturation granule neuron precursors in the DG during adult neurogenesis, in which Sox2 progenitors give rise to newborn neurons (Dcx+) differentiating into mature neurons (NeuN+). Our data, showing that Sox2+ cells in the DG were significantly decreased during acute colitis and unchanged during the chronic phase, ruled out the possibility that gut inflammation might promote brain tumor development, at least in our experimental setting. However, to draw more definitive conclusions on tumorigenesis, it might be worth exploring the effect of acute DSS treatment in future studies. Furthermore, possible inflammation-related tumor growth advantages were excluded based on the lack of differences in proliferation rate and macrophage infiltration in tumors from DSS-treated or untreated mice after labeling with Ki67 and Gfap.

In summary, we here reported the following: (i) depletion of a specific class of metabolites in fecal samples of DSS-treated mice related to CNS dysfunction; (ii) more severe hippocampal effects during acute vs. chronic DSS-induced colitis (cytokines, neurogenesis) (iii) local regional brain difference (hippocampus vs. cerebellum) in susceptibility to the neuroinflammation driven from DSS-induced gut inflammation (iv) lack of tumor-promoting effects of gut inflammation on MB development. Despite the metabolic perspective of this study being at the gut level, the comprehensive implications of the metabolic disorders should be taken from a systemic perspective due to the overall cascade of events that characterize this disease.

## 4. Materials and Methods

### 4.1. Animals

Wild-type or heterozygous mice lacking one *Ptch1* allele (Ptcneo67/+, termed *Ptch1^+/−^* throughout the text), generated through disruption of exons 6 and 7 in 129S2 ES cells [14] and maintained on the same C57BL/6 genetic background were used in this study. Mice were genotyped using polymerase chain reaction primers specific to the neo insert and wildtype regions as described previously [14]. Animals were housed under conventional conditions with food and water available ad libitum and a 12-h light-dark cycle.

### 4.2. Treatment with Dextran Sodium Sulfate (DSS)

For induction of acute colitis, male mice of 8 weeks of age were administered with a single cycle of 0.7% (*w*/*v*) DSS for 7 days followed by 3 or 7 days of DSS-free drinking water for molecular (n = 7) or histomorphological analyses (n = 10), respectively (Figure 1A). Corresponding age-matching untreated mice were sacrificed for molecular (n = 4) or histomorphological analyses (n = 5). Fecal samples were collected and immediately snap frozen for metabolomics analysis after one week of DSS treatment and at the end of experiment.

For induction of chronic colitis, male mice of 8 weeks of age were treated with 3 repeated DSS cycles of 7 days each followed by a recovery period of 14 days with DSS-free drinking water (Figure 1B). DSS-treated and age-matching untreated mice were sacrificed for molecular (UN, n = 6 and DSS, n = 6) and histological analysis (UN n = 7 and DSS n = 9). An extended protocol of chronic colitis was adopted for the evaluation of brain tumor development. In particular, *Ptch1^+/−^/*C57BL/6 mice of 8 weeks of age of mixed gender were treated with 5 DSS repeated cycles (n = 38) or left untreated (n = 37) (Figure 1C). Fecal samples were collected and immediately snap frozen for metabolomics analysis at the end of experiment.

### 4.3. Assessment of Acute and Chronic DSS-Induced Colitis

For both acute and chronic colitis, mice were daily checked for behavior, body weight, stool blood and consistency. The percentage of weight loss was calculated in relation to the starting weight using the following formula: [(Weight on day X − Initial weight)/Initial weight] × 100 [15]. At necropsy, colons removed and length measured from the anus to the top of the cecum. For histological analysis, fixed colonic tissues were embedded in paraffin, sectioned (4 µm thickness), mounted on glass slides and deparaffinized. Slices were stained using standard hematoxylin and eosin (H&E) techniques. Samples were analyzed by light microscopy and were scored according to the criteria of Maxwell et al. [15]. To evaluate the fibrotic score, fixed colonic tissues were embedded in paraffin, sectioned (6 µm), mounted on glass slides and stained using Masson’s Trichrome technique to highlight collagen fibers and proteoglycans. Samples, analyzed by light microscopy, were scored according to criteria of [82]. The fibrotic score was calculated by summing the scores relating to the amount of collagen and to the percentage of tissue involved. For the quantity of collagen; “0” is assigned to the colon sections without collagen increase; “1” for increase at the level of submucosa; “2” at the level of mucosa; “3” for both an increase in collagen in the muscularis mucosa and a thickening/disorganization of the same; “4” only for an increase in the muscularis propria and “5” for high level of disorganization of the muscularis propria and a thickening of the serosa. For the percentage of tissue involved, a score of “1” is assigned when the percentage ranges from 1 to 25% of the tissue; “2” when it ranges from 26 to 50%; “3” when it goes from 51 to 75% and “4” when it goes from 76 to 100%. The fibrotic score ranges from 0 to 9. Experiments were made in double-blind.

### 4.4. Collection of Brain Samples

Brains were removed and fixed immediately in a 10% (*w*/*v*) formalin solution or soaked in RNA later for subsequent hippocampus microdissection and storage in liquid nitrogen. For tumor quantification and histological analysis, brains were immediately fixed in a 10% (*w*/*v*) formalin solution. Samples were then embedded in paraffin wax according to standard techniques. Tumor incidence was expressed as the percentage of mice tumors in the total.

### 4.5. Immunohistochemistry

Sections (4 µm) of paraffin-embedded intestinal colonic and brain samples from mice were prepared following the standard protocol. Briefly, sections were dewaxed for 13 min at 90 °C with Heat Mediated Antigen Retrieval Solution (“HMARS”, Abcam, Germania, Germany) pH 6.0. Afterward, sections were washed in water for 5 min and peroxidases inhibited by incubation in 3% H_2_O_2_ for 10 min. Sections were treated with 5% bovine serum albumin (Santa Cruz Biotechnology, Santa Cruz, CA, USA) diluted in phosphate-buffered saline (PBS) for 30 min and incubated with the following primary antibodies: anti-Ki67 (IHC-00375, Bethyl, Montgomery, TX, USA, 1:500), anti-Dcx (ab18723, Abcam, 1:2000, Cambridge, UK), anti-Iba-1 (019-19741, Wako Pure Chemical Industries sections, Osaka, Japan, 1:500) and (ab1872315690, Abcam, 1:2000) were employed for immunostaining of the colon and the brain sections, anti-Gfap (Z0334, Dako, Germania, Germany, 1:500), anti-Sox2 (ab97959, Abcam, 1:500) diluted in PBS, overnight at 4 °C in a moist chamber. They were then washed in PBS, incubated for 1 h at room temperature with the secondary anti-rabbit antibody (Dako North America, Carpinteria, CA, USA) diluted 1:200 in PBS, and washed again. To visualize the antigen, sections were incubated with Vectastein Elite ABC (Vector Laboratories, Inc., Burlingame, CA, USA) for 30 min at room temperature, washed with PBS and incubated with Vector NovaRED Substrate Kit (Vector Laboratories, Inc., Burlingame, CA, USA) for 15 min at room temperature. Finally, samples were stained with H&E and analyzed by light microscopy using the software NIS-Elements BR 4.00.05 (Nikon Instruments Europe B.V.; Florence, Italy) at 4×, 10×, 20× and 100× magnification.

For adult neurogenesis, the brain of each mouse was cut sagittally to the midline and sections were collected starting at 500 μm from the midline. Images for quantification were taken using the imaging software NIS-Elements BR 4.00.05 (Nikon Instruments Europe B.V.; Florence, Italy). To standardize the counting area, cell quantification was performed on three non-overlapping serially collected sections per mouse, one from each hemisphere, representing the rostral/mid-hippocampus. The number of positive cells in each section was expressed per mm of the SGZ length. NSCs were counted based on criteria including SGZ localization, positive labeling and morphology.

For tumor phenotyping, a subset of brain sections from *Ptch1^+/−^* mice with MB developed during the temporal window of DSS treatments were immunostained for Gfap and Ki67 and specific regions of interest (hippocampus, VIII lobule of cerebellum and tumor tissue) and analyzed by HistoQuest software (Tissue Gnostics, Wien, Austria—for automatic color separation and quantification). All experiments were analyzed in blind.

### 4.6. Real-Time qPCR

Total RNA was isolated from mouse colonic and hippocampal tissues using the RNeasy kit (QiaGenGmbH, Hilden, Germany), and 1 µg of total RNA was reverse transcribed by an IScriptTM cDNA Synthesis Kit (BioRad, Hercules, CA, USA). The qPCR amplifications were obtained by a BioRad CFX96 TouchTM Real-Time PCR Detection System using SsoAdvanced Universal SYBR Green super Mix (BioRad). The following primers were used: *Rpl32* fwd primer 5′-TGTGCAACAAATCTTACTGTGCT-3′, rvs primer 5′-TGCACACAAGCCATCTACTCA-3′; *Il*-*6* fwd primer 5′-CAAGTCGGAGGCTTAATTACACATG-3′, rvs primer 5′-TGCCATTGCACAACTCTTTTCT-3′, *Il*-*1β* fwd primer 5′-CAGGCAGTATCACTCATTG-3′, rvs primer 5′-CGTTGCTTGGTTCTCCTTGT-3′; *S*-*100* fwd primer 5′-TCATGGAGGACCTGGACACA-3′, rvs primer 5′-CAGCATCATACACTCCTCAAAGCT-3′; *Tgf*-*β3* fwd primer 5′-AGCTGCCAAAGTGTG-3′, rvs primer 5′-CAGGGGCTTGGTGAA-3′; *Smad*-3 fwd primer 5′-GCCCTCCTAGCTCAGTCTGTCA-3′, rvs primer 5′-ATGCGCGTCAGCTGGTAGA-3′; *iNOS* 5′-GGTGAAGGGACTGAGCTGTTAGA-3′, rvs primer 5′-GTTCTCCGTTCTCTTGCAGTTGA-3′. *Rpl32* expression level was used to normalize mRNA expression of target genes. The quantity of mRNA relative to the reference gene was calculated by the 2^−ΔCT^ method.

### 4.7. Processing of Stool Samples and Metabolomics

For semipolar analysis, the sample preparations were according to previously published methods with little modifications [83,84]. More specifically, a pool of 2 frozen stool samples from 2 independent mice (total dry weigh 15–25 mg) from the different experimental groups [UN-Acute n = 5; UN-Chronic n = 7; DSS-acute, at the end (n = 5) or one week after DSS treatment (n = 5) and DSS-chronic (n = 8)] were mixed and shacked for two times in 750 μL of aqueous solution with 75% of methanol, 0.1% of formic acid and 1 μg/mL of formonetin as internal control for 20 min at room temperature. The mixture was then vortexed for 3 min followed by centrifugation at 15,000× *g* for 15 min. The suspension (600 μL) was used for metabolic profiling. For the LC-ESI-MS analysis, samples were transferred to HPLC tubes and an aliquot of 3 μL was injected for the analysis. Liquid chromatography coupled to high-resolution mass spectrometry conditions was previously reported [85]. Untargeted metabolomics was performed using the Compound Discovered software (Thermo Fisher Scientific, Waltham, MA, USA), which performs chromatogram alignment, peak picking and public database (e.g., ChemSpider, KEGG, Metabolika) querying based on accurate masses (*m*/*z*). After chromatogram alignment and retrieval of all the detected frames (e.g., ions), the data generated were normalized with respect to internal standards. Differentially accumulated metabolites (DAMs) were detected by Student’s *t*-test. A *p*-value ≤ 0.05 was considered to indicate a statistically significant difference. For metabolite identification, a custom pipeline annotation was used by using, at first glance, Compound Discoverer software followed by manual curation using the Metlin database (https://metlin.scripps.edu/, accessed on 3 August 2022). Tentative identifications were validated by comparing chromatographic and spectral properties with authentic standards (when available) and reference spectra, in-house database, literature data and based on the *m*/*z* accurate masses, as reported in the Pubchem database (http://pubchem.ncbi.nlm.nih.gov/, accessed on 3 August 2022) for monoisotopic mass identification, or on the Metabolomics Fiehn Lab Mass Spectrometry Adduct Calculator(http://fiehnlab.ucdavis.edu/staff/kind/Metabolomics/MS-Adduct-Calculator/, accessed on 3 August 2022) for adduction detection, subsequently confirmed by MS/MS fragmentation.

Principal component analysis (PCA) untargeted metabolomes were performed using the Past4 software [86]. MetaboAnalyst 5.0 (https://www.metaboanalyst.ca/, accessed on 5 August 2022) combined with manual curation to validate and integrate when needed, the assignment of each metabolite to the corresponding pathway and chemical structure was used to analyze the super class, pathways and signal diseases associated with the differential metabolites identified by comparing the UN and the DSS groups and acute DSS-treated and the chronic DSS-treated groups.

### 4.8. Statistics

Data were given as mean ± standard error of the mean (SEM). All statistical analyses were carried out using GraphPad InStat software (GraphPad, San Diego, CA, USA). The Kolmogorov–Smirnov test was used to assess whether data were sampled from populations following the Gaussian distribution. Comparisons between groups were performed using the non-parametric Mann-Whitney test (significance taken as *p* < 0.05).

## 5. Conclusions

Whereas it is well-established that gut inflammation is associated with depression-like behaviors in IBD patients, our understanding of the pathological mechanisms of brain dysfunction during disease progression is still rather limited. We established a functional connection between the presence of colon inflammation, metabolic alterations, neuroinflammation and alteration of hippocampal neurogenesis. On the other hand, we showed that neuroinflammation did not involve the cerebellum region, and consequently, the rate of MB tumorigenesis was not influenced by the presence of gut inflammation driven by DSS treatment. An animal model of IBD might help to elucidate the underlying mechanisms of neurobiological alterations in the CNS and help to identify targets for the development of better therapeutic options for the treatment of IBD-associated comorbidities such as depression.

## Figures and Tables

**Figure 1 ijms-23-11495-f001:**
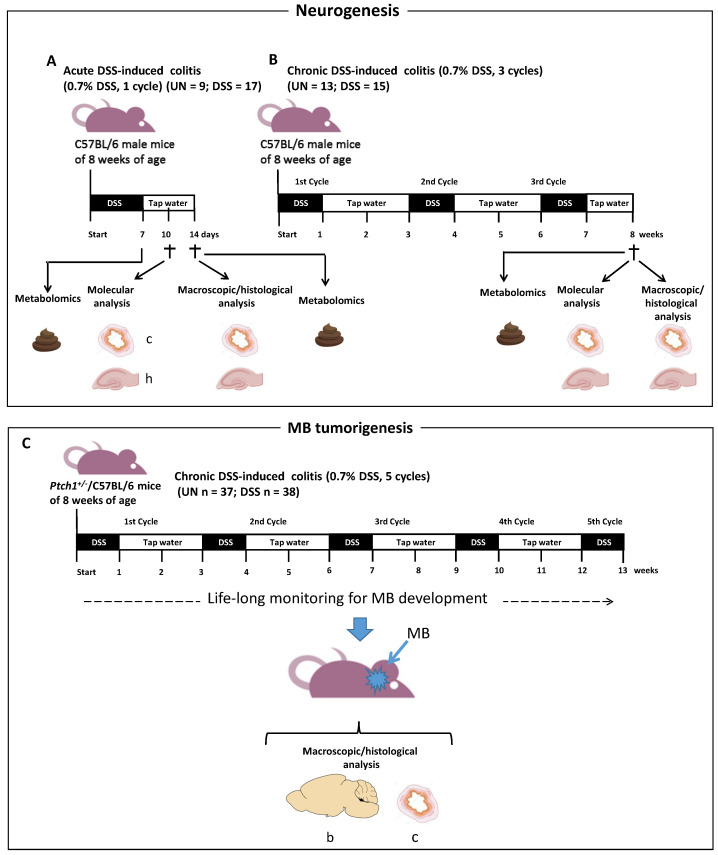
Treatment schemes, size of the groups and endpoints. UN: untreated group. DSS: sodium dextran sulfate-induced colitis group. (**A**) Induction of acute DSS-induced colitis (**B**) Induction of chronic DSS colitis. (**C**) Extended chronic DSS-induced colitis for tumorigenesis. †—time points of sacrifice, C–colon, h–hippocampus, b–brain, s–stool. The number of mice used for each analysis is indicated in the corresponding sections (n).

**Figure 2 ijms-23-11495-f002:**
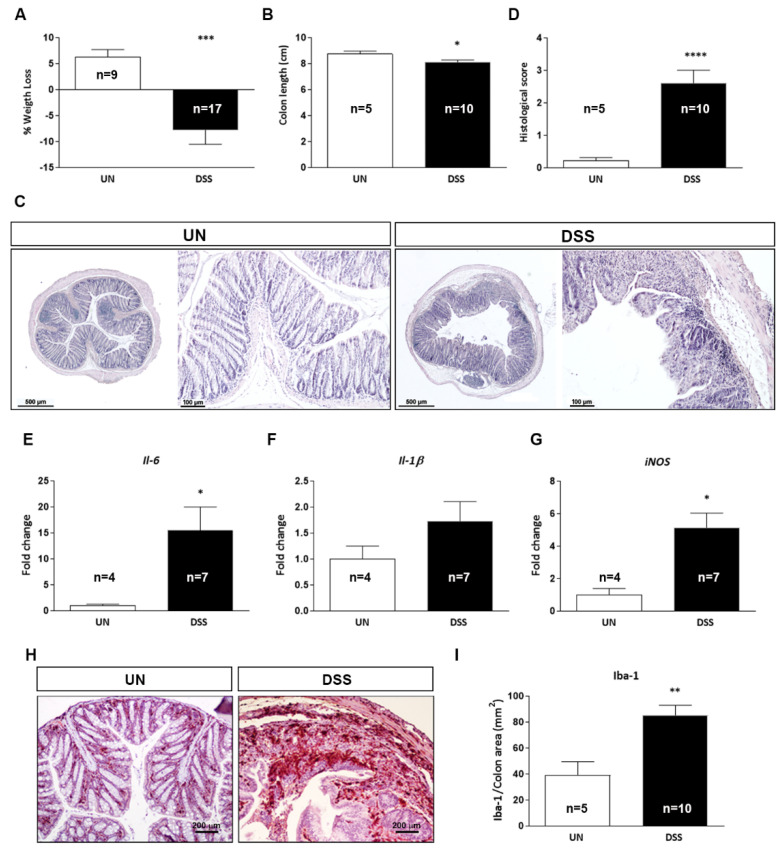
Macroscopic, molecular and histological analyses in the colon of C57BL/6 mice with DSS-induced acute colitis. Body weight loss (**A**) and colon length (**B**). Representative colon sections from DSS-treated and untreated mice (**C**) and relative histological score (**D**). mRNA expression levels of proinflammatory cytokines (**E**) *Il*-*6*, (**F**) *Il*-*1β* and of the oxidative stress marker (**G**) *iNOS* were analyzed by qPCR. Representative images of microglial marker Iba-1 expression by IHC (**H**) and relative quantification (**I**). UN: untreated group; DSS: sodium dextran sulfate treated group. The number of mice used per test is indicated in the graphs (n). Data are reported as mean ± SEM, * *p* < 0.05; ** *p* < 0.01; *** *p* < 0.001; **** *p* < 0.0001 for comparison with controls.

**Figure 3 ijms-23-11495-f003:**
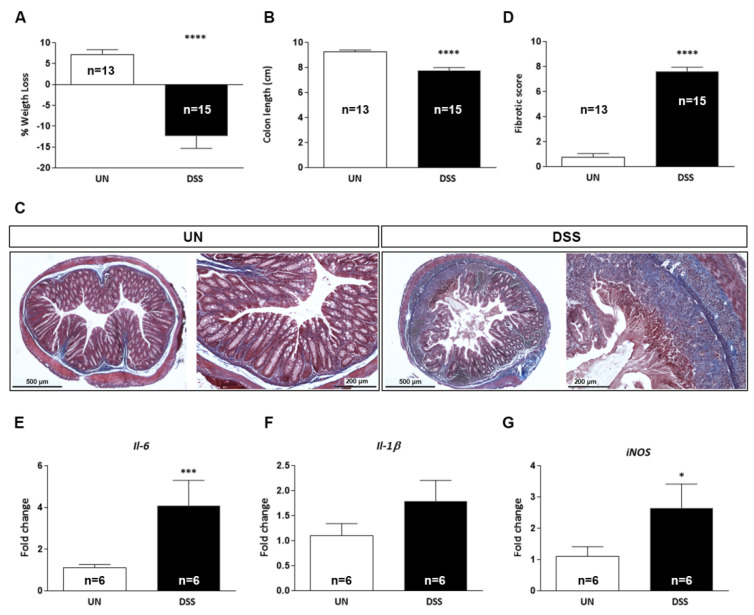
Macroscopic, molecular and histological analyses in the colon of C57BL/6 mice with DSS-induced chronic colitis. Following parameters were analyzed: body weight loss (**A**), colon length (**B**), Masson’s trichrome-stained sections of colon (**C**), fibrotic score (**D**). mRNA expression levels of proinflammatory cytokines (**E**) *Il*-*6*, (**F**) *Il*-*1β* and of the oxidative stress marker (**G**) *iNOS* were analyzed by qPCR. UN: untreated group; DSS: sodium dextran sulfate treated group. The number of mice used per test is indicated in the graphs (n). Data are reported as mean ± SEM, * *p* < 0.05; *** *p* < 0.001; **** *p* < 0.0001 for comparison with controls.

**Figure 4 ijms-23-11495-f004:**
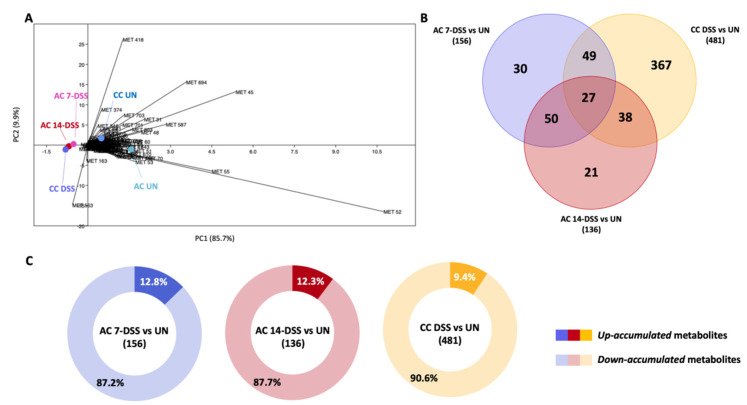
(**A**) Principal component analysis (PCA) of the profiling data from the fecal metabolome. (**B**) Venn diagram showing the common and unique metabolites biologically relevant in each comparison. (**C**) Circular diagram summarizing the percentage of up and down-regulated fecal metabolites found to be biologically relevant in each comparison. AC-7-DSS and AC-14-DSS: Mice with acute colitis at 7 or 14 days from the start of DSS treatment. CC DSS: chronically DSS-treated mice. UN: untreated mice.

**Figure 5 ijms-23-11495-f005:**
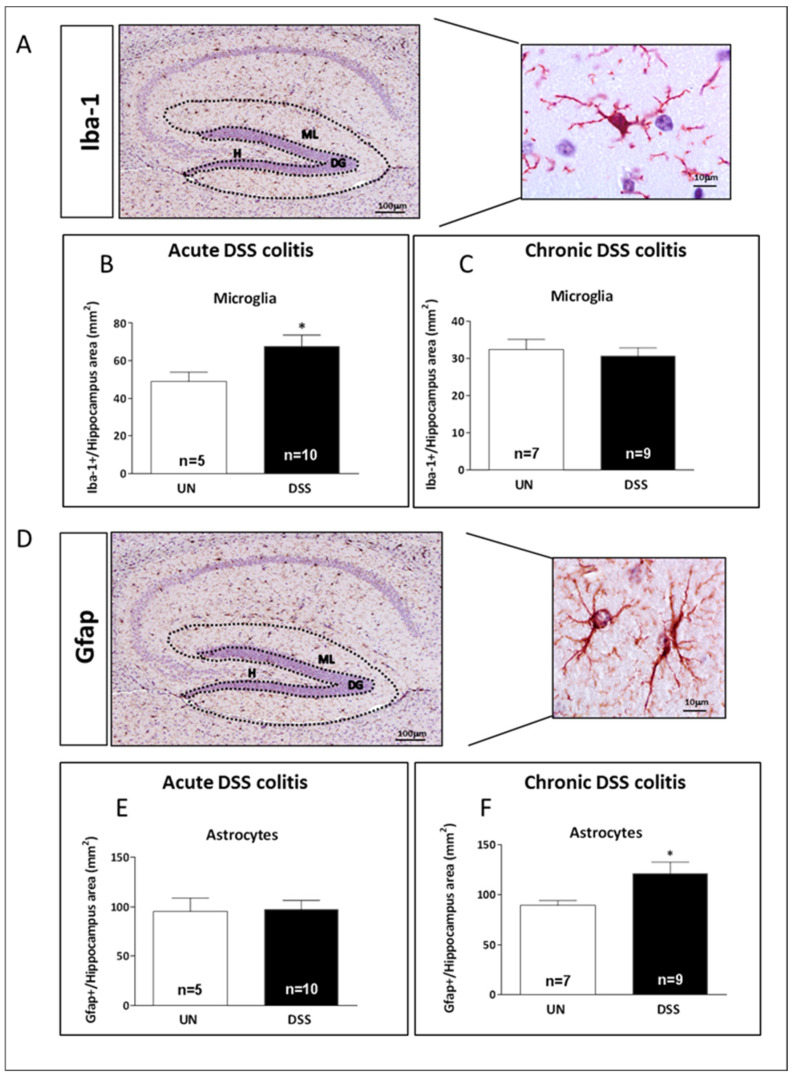
Neuroinflammatory responses in the hippocampus after acute and chronic DSS-induced gut colitis. (**A**) Representative image of Iba-1 immunostaining in specific regions (dashed lines). Quantification of Iba-1 expression level in brain sections of (**B**) acute and (**C**) chronic DSS-treated mice. (**D**) Representative images of Gfap immunostaining in specific regions (dashed lines). Quantification of Gfap expression level in brain sections of (**E**) acute and (**F**) chronic DSS-treated mice. H: hilus, ML: molecular layer; DG: dentate gyrus. The number of mice used per test is indicated in the graphs (n). Data are reported as mean ± SEM, * *p* < 0.05 for comparison with controls.

**Figure 6 ijms-23-11495-f006:**
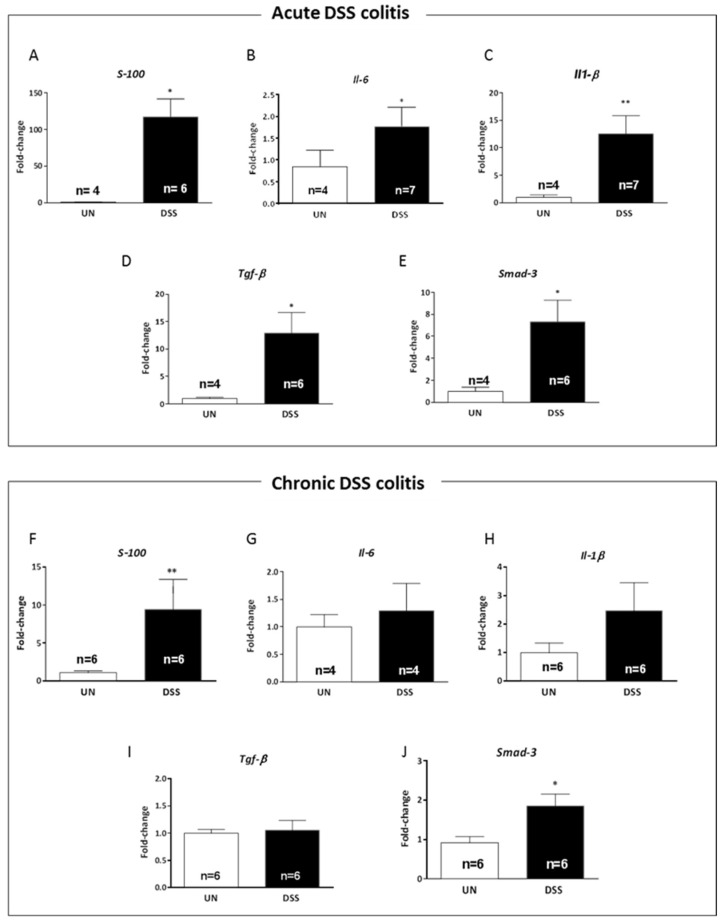
Cytokines expression profiles in the hippocampus following acute or chronic DSS colitis. (**A**–**E**) mRNA expression levels of proinflammatory cytokines (**A**) *S*-*100*, (**B**) *Il*-*6*, (**C**) *Il*-*1β*, (**D**) *Tgf*-*β* and (E) *Smad*-*3* after acute DSS treatment and (**F**–**J**) mRNA expression levels of (**F**) *S*-*100*, (**G**) *Il*-*6*, (**H**) *Il*-*1β*, (**I**) *Tgf*-*β* and (**J**) *Smad*-*3* after chronic DSS treatments. UN: untreated group; DSS: sodium dextran sulfate-treated group. The number of mice used per test is indicated in the graphs (n). Data are reported as mean ± SEM, * *p* < 0.05; ** *p* < 0.01 for comparison with controls.

**Figure 7 ijms-23-11495-f007:**
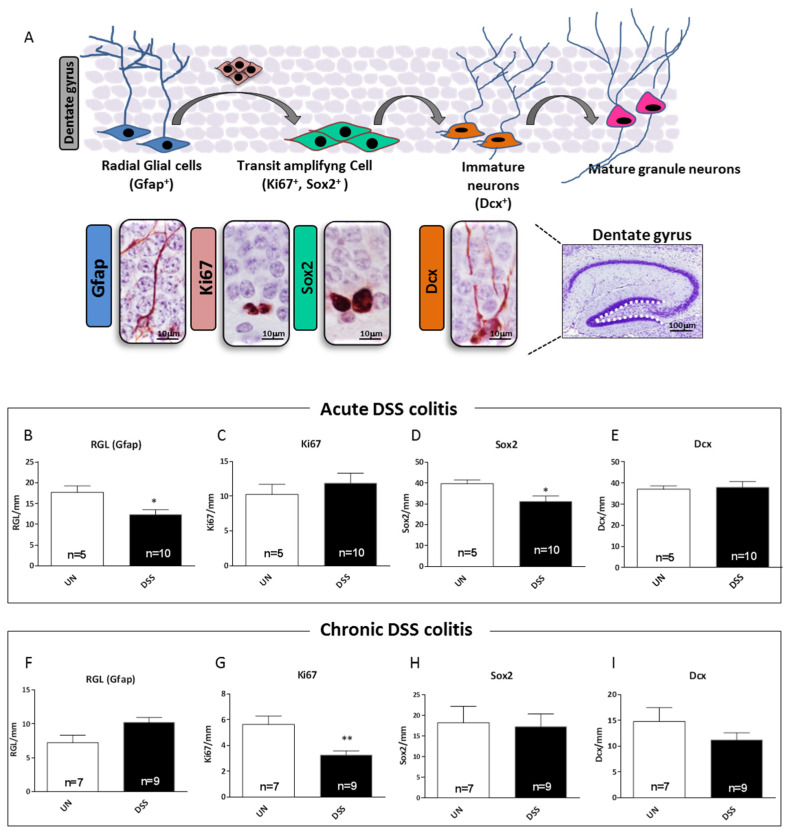
Hippocampal neurogenesis in mice with acute and chronic DSS-induced colitis. (**A**) Schematic representation of adult neurogenesis in the DG and relative stage-specific markers. Representative images of cell labeled by glial fibrillary acidic protein, Gfap, sex determining region Y (SRY) box 2, Sox2, Ki67 and doublecortin, Dcx. The white line in the DG delineates the area in which all the analyses were carried out. (**B**–**E**) Relative quantification for stage-specific neurogenesis markers after DSS-acute treatment. (**F**–**I**) Relative quantification for stage-specific neurogenesis markers after DSS-chronic treatment. UN: untreated group; DSS: sodium dextran sulfate-treated group. The number of mice used per test is indicated in the graphs (n). Data are reported as mean ± SEM * *p* < 0.05; ** *p* < 0.01 for comparison with untreated controls.

**Figure 8 ijms-23-11495-f008:**
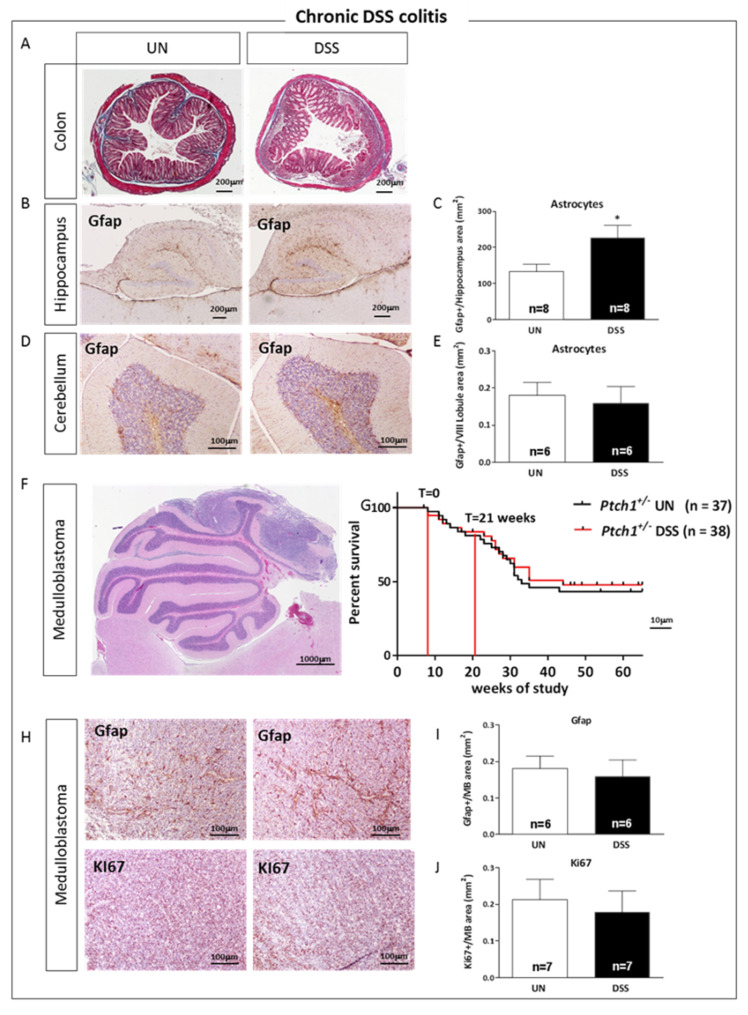
Gut inflammation and brain tumorigenesis relationship in *Ptch1^+/−^*/C57BL6 mice. (**A**) Masson’s trichrome-stained sections of colon from UN- and DSS-treated mice. (**B**–**E**) Representative images of immunohistochemistry with Gfap antibody in the hippocampus and cerebellum (VIII lobule) sections from UN and DSS mice and their relative quantification. (**F**) Representative images of hematoxylin/eosin-labeled MB sections. (**G**) Kaplan–Meier overall survival curves of UN- and DSS-treated mice; T0 and T21 respectively mark the start of DSS treatment (8 weeks of age) and the end of the treatment (T21). (**H**–**J**) Representative images of immunostaining with Gfap and Ki67 antibodies in MB tissue sections and their relative quantification. UN: untreated group; DSS: sodium dextran sulfate-treated group. The number of mice used per test is indicated in the graphs (n). Data are reported as mean ± SEM, * *p* < 0.05 for comparison with controls.

**Table 1 ijms-23-11495-t001:** List of differentially accumulated known metabolites in fecal samples of DSS acute and chronic induced colitis (AC-7, AC-14 and CC) vs. untreated group.

Class	Metabolite	Formula	Adduct	FC DSS/UN		
				AC-7	AC-14	CC
Amines	N-acetyloctopamine	C10H13NO3	[M-H]−1	−0.11	−0.27 *	−0.27 *
Amino Acids, Peptides and Proteins	Thyronine	C15H15NO4	[M+H]+1	−1.08	−1.50	−1.16 *
	L-SerineO-sulfate	C3H7NO6S	[M+FA-H]−	−1.84 *	−0.95	−0.42
	Val-Val	C10H20 N2O3	[M+H]+1	−1.13 *	−0.03	0.81
	3,5-Diiodo-L-tyrosine	C8H9 I2 NO	[M+H]+1	−1.46 *	−0.71	−0.77
	N-lauroyl glycine	C14H27 NO3	[M+H]+1	−0.86	−1.02	−1.69 *
	D-Aspartic acid	C4H7NO4	[M+H]+1	−0.42	−0.04	−0.87 *
Fatty Acyls [FA]	(S)-N-(1-amino-1-oxo-3-phenylpropan-2-yl)oleamide	C27H45NO3	[M+H-H2O]+	−0.95	−1.62	−2.10 *
	methyl 8-[2-(2-formyl-vinyl)-3-hydroxy-5-oxo-cyclopentyl]-octanoate	C17H26O5	[M+H]+1	−0.29	−0.48 *	−0.37 *
	Pimelic acid	C7H12O4	[M-H]−1	−0.68	−0.61	−1.79 *
	Suberic acid	C8H14O4	[M-H]−1	−0.47	−0.32	−2.08 *
	Azelaic acid	C9H16O4	[M-H]−1	−0.27	−0.19	−1.79 *
	Oleoyl Ethanolamide	C20H39NO2	[M+H]+1	−2.14 *	−1.32	−1.48 *
	Linoleyl carnitine	C25H45 NO4	[M+H]+1	−2.73 *	−1.93	0.13
	Dihydrozeatin-O-glucoside-like	C16H25 N5O6	[M-H]−1	−2.19	0.81	−1.31 *
	15-Deoxy-Δ12.14-prostaglandin J2-2-glycerol ester	C23H34O5	[M+H]+1	−0.63	−0.98	−2.18 *
	9Z-Eicosenoic acid	C14H30O8	[M+NH4]+1	−0.68	−1.06	−0.23 *
	Hexadecanedioic acid Mono-L-carnitine ester	C23H43 NO6	[M+H]+1	−0.81	−0.91	−2.17 *
	Palmitoleoyl-EA	C18H35 NO2	[M+H]+1	−1.42	−1.28	−1.62 *
	Oleamide	C18H35 NO	[M+H]+1	−1.33	−1.64	−1.15 *
	Octadecanamide	C18H37 NO	[M+H]+1	−1.42	−1.50 *	−1.19 *
Glycerophospholipids	LysoPC(18:1(9Z))	C26H52NO7 P	[M+H]+1	0.16	−0.27	−1.13 *
Phospholipids	PI(O-16:0/0:0)	C32H44O7	[M-H]−1	−0.35	−2.13	−1.60 *
Prenol Lipids	3.4-didehydro-retinoate	C20H26O2	[M-H]−1	−0.05	0.05	−0.37 *
	9-cis-Retinoic acid	C20H28O2	[M-H]−1	−0.47	−0.12	−2.00 *
	Alpha-Tocotrienol	C29H44O2	[M+H]+1	−0.86	−1.19	−1.75 *
	Ecabet-like	C20H28O5S	[M-H]−1	−3.21 *	−1.82	−1.34 *
Sterol lipids	ST 28:2;O7	C28H46O7	[M+Na]+	−1.47	−2.05 *	−2.41 *
	Abrusoside A-like	C36H54O10	[M-H]−1	−1.67 *	−1.12	−0.67
	1-a.24R.25-Trihydroxyvitamin D2	C28H44O4	[M+CH3COO]−	0.74	1.45	−0.89
	1α.25-dihydroxy-26.27-dimethyl-20.21-methano-23-oxavitamin D3/1α.25-dihydroxy-26.27-dimethyl-20.21-methano-23-oxacholecalciferol	C29H46O4	[M+FA-H]−	−1.76	−1.61	−1.93 *
	11b-hydroxyprogesterone	C20H28O4	[M-H]−1	−2.17	−0.36	−1.95 *
	16alpha.17-Isopropylidenedioxy-6alPha-methylprogesterone	C25H36O4	[M+CH3COO]−	−0.60 *	−0.58	−0.89 *
Sterol lipids/Bile acid	Lithocholic acid glycine conjugate	C26H43 NO4	[M+H]+1	−0.88	−1.14	−1.61 *
Phenylpropanoids and polyketides	Afrormosin-like	C17H14O5	[M+H]+1	−0.56	−0.95	−1.51 *
	Bergaptol-like	C11H6O4	[M+H-H2O]+1	−1.04 *	−1.16 *	−0.44
Organoheterocyclic compounds	5-(2-Hydroxyethyl)-4-methylthiazole	C6H9NOS	[M+H]+1	−0.90	−0.71	−1.13 *
	Thiamine	C12H16N4OS	[M+2H]+2	−0.70	−0.32	−1.03 *
	Iprodione-like	C13H13Cl2N3O3	[M-H]−1	−0.00	0.28	−1.27 *
	Mesobilirubinogen	C33H44N4O6	[M-H]−1	−0.64	−1.57	−1.49 *
Organic Nitrogen compounds	N,N-Dimethylsphing-4-enine	C20H41NO2	[M+H]+1	−1.20	−2.32 *	−1.18 *
Organic acids and derivative	Tazobactam-like	C10H12N4O5S	[M+H-H2O]+1	0.07	0.26	−1.21 *
	4-Hydroxyphenylacetylglycine	C10H11NO4	[M-H]−1	1.66 *	1.57	0.66
Nucleosides, Nucleotides and analogues	5’-Deoxyadenosine	C10H13N5O3	[M+H]+1	−1.12 *	−1.55 *	−0.89
	4-Amino-5-hydroxymethyl-2-methylpyrimidine	C6H9 N3O	[M+H]+1	−1.45	−1.13	−1.03 *
Benzenoids	Phenylethylamine	C14H27 NSi2	[M+H]+1	−1.06	−1.34	−1.45 *
	Metoclopramide-like	C14H22ClN3O2	[M+H]+1	−0.94	−1.03	−2.07 *
	Butylparaben-like	C11H14O3	[M-H]−1	−1.24	−0.82	−1.50 *

* Statistically significant changes.

## Data Availability

Metabolomics data generated within this study have been deposited into the NIH Common Fund’s National Metabolomics Data Repository (NMDR), the Metabolomics Workbench, https://www.metabolomicsworkbench.org. Project ID PR001456. The data can be accessed directly via its Project DOI: 10.21228/M8HQ5P, without restrictions.

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
