# Peer review of "Gut–Brain Axis: Insights from Hippocampal Neurogenesis and Brain Tumor Development in a Mouse Model of Experimental Colitis Induced by Dextran Sodium Sulfate"

_ijms, 2022, doi:10.3390/ijms231911495_

Round 1
Reviewer 1 Report
Many thanks for your invitation to review the manuscript titled: Gut-Brain Axis: ‘’Insights from Hippocampal Neurogenesis and Brain Tumor Development in a Mouse Model of Experimental Colitis Induced by Dextran Sodium Sulfate’’
Overall this is a well-designed and well executed study with novel findings, which deserves to be published .
I only have some minor points to suggest for revision:
1. I would suggest to the authors to have the text reviewed by a person fluent in English, as there are quite a few syntax/grammar errors. For instance, the text in lines 49-51 could be edited to: ‘For example, it has recently been reported that IBD patients exhibit double the rate of depression compared to healthy controls, with this prevalence being heightened during active IBD [2]’
2. Similarly, in Line 65 please consider rephrasing the sentence ‘’either acute or chronic’’ to ‘both acute and chronic’
3. In regards with the text in Lines 369-373 and Lines 380-387: It might be better if these parts are either ‘moved’ in the introduction section, or incorporated and better ‘linked’ later in the discussion with the study’s findings
4. Similarly the paragraph in lines 424-435 could be incorporated in the discussion in a better manner and linked to the study’s findings, as this would give the discussion a better flow
5. Finally, I also suggest that in the discussion, at the end of each section (i.e acute colitis, chronic colitis, oncogenesis), the authors add a small paragraph summarising their findings and relevant conclusions . This would make the discussion much easier to read
Author Response
Reviewer 1 Overall, this is a well-designed and well executed study with novel findings, which deserves to be published. I only have some minor points to suggest for revision: 1. I would suggest to the authors to have the text reviewed by a person fluent in English, as there are quite a few syntax/grammar errors. For instance, the text in lines 49-51 could be edited to: ‘For example, it has recently been reported that IBD patients exhibit double the rate of depression compared to healthy controls, with this prevalence being heightened during active IBD [2]’ 2. Similarly, in Line 65 please consider rephrasing the sentence ‘’either acute or chronic’’ to ‘both acute and chronic’
R: 1 and 2. The suggested changes in points 1 and 2 have been introduced in the revised text.
3. In regards with the text in Lines 369-373 and Lines 380-387: It might be better if these parts are either ‘moved’ in the introduction section, or incorporated and better ‘linked’ later in the discussion with the study’s findings
R: Thank you for these suggestions, we moved lines 380-387 later in the discussion.
4. Similarly the paragraph in lines 424-435 could be incorporated in the discussion in a better manner and linked to the study’s findings, as this would give the discussion a better flow
R: We revised the text to better integrate this paragraph.
5. Finally, I also suggest that in the discussion, at the end of each section (i.e acute colitis, chronic colitis, oncogenesis), the authors add a small paragraph summarising their findings and relevant conclusions. This would make the discussion much easier to read
R: Highlights recapitulating the major findings and conclusions of the study are presented at the end of the discussion and Conclusions are also presented in Paragraph 5. We are afraid that the addition of summaries at the end of each section might become repetitive and redundant.
Reviewer 2 Report
In their manuscript, Vitali et al has used acute and chronic mouse models of chemical colitis with dextran sodium sulfate (DSS) with a goal to elucidate the crosstalk between the gut-brain axis, focusing on the consequences of intestinal inflammation on neuroinflammation and oncogenesis. The authors successfully recapitulated the markers of intestinal inflammation in both acute and chronic DSS colitis in mice like body weight loss, colon shortening, histological damage, colonic macrophage infiltration and increased expression of IL-6 and iNOS. The novel observations were the fecal metabolic changes the appearance of proinflammatory markers in the hippocampus of acutely DSS-treated mice (Il-6, Il-1b, S-100, Tgf-b and Smad-3), along with microgliosis. They found that chronic DSS treatment also induced neuroinflammation in the hippocampus as indicated by astrocytes activation. Further analysis of stage-specific neurogenesis markers indicated that acute and chronic DSS treatments impaired adult hippocampal neurogenesis. To correlate gut-related inflammation and brain cancer, the authors also utilized medulloblastoma (MB) mouse model (Ptch1+/-/C57BL/6 mice). However no differences were observed in MB development between untreated and DSS-treated mice. The authors conclude that acute and chronic intestinal inflammation can significantly impair brain homeostasis via impairing hippocampal neurogenesis, but not MB oncogenesis.
Major comments.
11. The DSS dose used in the study is 0.7%. It is seen from supplemental data files that commonly used DSS concentrations of 2.5% caused 100% mortality in mice. This is unusual as DSS between 3 and 5 % are used to model colitis in C57BL6 mice without a 100% mortality. But if the mice were otherwise healthy and without infections, then it might be attributable to a specific batch of DSS.
22. MB development in Ptch1+/- /C57BL/6 mice occurs naturally between 10 and 22 weeks of mouse age. Since the goal of the experiment is to identify how gut inflammation can regulate or accelerate the onset of oncogenesis, it would have been appropriate to treat the mice in their early ages (between 6 to 8 weeks of age with acute DSS model) and then see if they developed tumor faster/ or at an earlier age than their counterparts that is the normal Ptch1+/- 339 /C57BL/6 mice, where the tumor onset is expected at 10 weeks of age. It appears that the authors, by using a chronic DSS protocol (5 repeated DSS cycles, each alternating with 2 weeks with tap water) spanning 8 to 21 weeks of mouse age, might have missed the difference. Here again, the DSS dose could be a crucial factor as well.
33. Did authors look at TNF levels in their MB model? This is important as MB cell proliferation is highly stimulated by complement C3a triggered astrocyte activation and TNF-α production.
44. Also did authors compare the fecal metabolomes from MB model in DSS-treated versus untreated mice? This might offer some directions regarding the pathways involved.
55. Very clear schematic diagrams for the experimental methods deserve special mention. Could include the DSS concentration in the diagrams as well.
Minor comments
1. Check for spelling in general, line 580- change to “cytokines”.
Author Response
Reviewer 2 In their manuscript, Vitali et al has used acute and chronic mouse models of chemical colitis with dextran sodium sulfate (DSS) with a goal to elucidate the crosstalk between the gut-brain axis, focusing on the consequences of intestinal inflammation on neuroinflammation and oncogenesis. The authors successfully recapitulated the markers of intestinal inflammation in both acute and chronic DSS colitis in mice like body weight loss, colon shortening, histological damage, colonic macrophage infiltration and increased expression of IL-6 and iNOS. The novel observations were the fecal metabolic changes the appearance of proinflammatory markers in the hippocampus of acutely DSS-treated mice (Il-6, Il-1, S-100, Tgf- and Smad-3), along with microgliosis. They found that chronic DSS treatment also induced neuroinflammation in the hippocampus as indicated by astrocytes activation. Further analysis of stage-specific neurogenesis markers indicated that acute and chronic DSS treatments impaired adult hippocampal neurogenesis. To correlate gut-related inflammation and brain cancer, the authors also utilized medulloblastoma (MB) mouse model (Ptch1+/-/C57BL/6 mice). However no differences were observed in MB development between untreated and DSS-treated mice. The authors conclude that acute and chronic intestinal inflammation can significantly impair brain homeostasis via impairing hippocampal neurogenesis, but not MB oncogenesis. Major comments. 1. The DSS dose used in the study is 0.7%. It is seen from supplemental data files that commonly used DSS concentrations of 2.5% caused 100% mortality in mice. This is unusual as DSS between 3 and 5 % are used to model colitis in C57BL6 mice without a 100% mortality. But if the mice were otherwise healthy and without infections, then it might be attributable to a specific batch of DSS.
R: The authors completely agree with this comment, and they considered the concentration-related differences with the majority of the published studies due to the brand/batch of DSS. In fact, the pilot study for the identification of the appropriate DSS concentration has been set just in consideration of this.
2. MB development in Ptch1+/- /C57BL/6 mice occurs naturally between 10 and 22 weeks of mouse age. Since the goal of the experiment is to identify how gut inflammation can regulate or accelerate the onset of oncogenesis, it would have been appropriate to treat the mice in their early ages (between 6 to 8 weeks of age with acute DSS model) and then see if they developed tumor faster/ or at an earlier age than their counterparts that is the normal Ptch1+/- 339 /C57BL/6 mice, where the tumor onset is expected at 10 weeks of age. It appears that the authors, by using a chronic DSS protocol (5 repeated DSS cycles, each alternating with 2 weeks with tap water) spanning 8 to 21 weeks of mouse age, might have missed the difference. Here again, the DSS dose could be a crucial factor as well.
R: The authors agree that DSS dose, starting time and duration of the treatment are all important determinants in the outcome of tumorigenesis experiment. However, the rationale behind the choice of long DSS exposure, spanning most the tumor latency, was to increase the chance of picking up DSS-related tumor difference. To keep these considerations into account the authors added the following sentence “However, to draw more definitive conclusions on tumorigenesis it might be worth exploring the effect of acute DSS treatment in future studies” in the Discussion.
In addition, treating with DSS young mice of 6 weeks of age in which body weight is frequently below 20gr would have constitute threatening risk for mortality.
3. Did authors look at TNF levels in their MB model? This is important as MB cell proliferation is highly stimulated by complement C3a triggered astrocyte activation and TNF-α production.
R: Given the apparent lack of effect of gut inflammation on MB tumorigenesis and in the absence of DSS-driven differences in MB cell proliferation (Ki67 in Fig. 8H) we have not addressed this point.
4. Also did authors compare the fecal metabolomes from MB model in DSS-treated versus untreated mice? This might offer some directions regarding the pathways involved.
R: As for point 3, given the lack of influence of gut inflammation on tumorigenesis in our current settings, and in consideration of the high cost of metabolomics, we skip the analyses of the fecal metabolome in DSS-treated vs untreated Ptch1+/- mice.
However, future experiments to address the effect of acute DSS treatment on MB development, in which mice will be not monitored for the whole life-time for tumor development, but sacrificed at a given time, even without MB signs, and evaluated for MB may be foreseen for a new study. These animals could represent a more homogeneous cohort for mechanistic investigations, including TNF expression levels and evaluation of fecal metabolic changes.
5. Very clear schematic diagrams for the experimental methods deserve special mention. Could include the DSS concentration in the diagrams as well.
R: Thank you for your kind appreciation, the DSS concentration has been included in the scheme.
Minor comments
1. Check for spelling in general, line 580- change to “cytokines”.
R: The misspelling has been corrected
